# Dose-related association between radiation exposure from computed tomography (CT) scans during trauma hospitalizations and subsequent risk of developing new-onset cancers
Lai Kin Yaw[1,2], Swithin Song[3] & Kwok Ming Ho [●][4,5,6] [✉]

## Abstract

**Background** The association between radiation dose from Computed Tomography (CT) and subsequent cancer risk in adults remains poorly defined.
**Methods** We conducted a statewide cohort study to examine the relationship between CT-related radiation exposure — measured by dose-length-product (DLP) — and cancer outcomes among adult trauma patients in Western Australia from 2004 to 2020. Patients with a documented cancer diagnosis within five years prior to trauma were excluded.
**Results** After excluding patients with missing smoking data ($n = 12,690$), 2662 patients (17.3%) are included in the primary analysis. The cohort is predominantly male (75.8%), with a median age of 41 years (IQR: 27–58) and a median Injury Severity Score (ISS) of 17 (IQR: 16–22). Over a median follow-up of 5.9 years (IQR: 4.0-7.9), 374 patients (14.0%) died, including 21 cancer-related deaths (0.8%), accounting for 5.6% of all deaths. During the index trauma admission, patients underwent a median of 6 X-rays (IQR: 3-12) and 3 CT scans (IQR:1-5) with a median DLP of 1,941 mGy*cm (IQR: 637-3,388). DLP and absorbed radiation dose are significantly correlated with injury severity (Pearson $r = 0.209$ and 0.265, respectively; both $p = 0.001$). Radiation exposure is significantly associated with increased risk of new-onset cancer (adjusted hazard ratio [aHR]: 1.08 per 1,000 mGy*cm increment in DLP; 95%CI: 1.01-1.16; $p = 0.042$) and cancer-related mortality (aHR 3.35 for those exposed to >5000 mGy*cm; 95%CI: 1.20-9.38; $p = 0.021$). These findings are consistent in a larger cohort of 15,352 patients after multiple imputation for missing smoking data.
**Conclusions** CT-related radiation exposure during trauma hospitalizations is associated with a dose-dependent increase in the risk of subsequent cancer incidence and mortality.

## Plain language summary

Medical imaging, especially CT (Computed Tomography) scans, uses X-rays and computer technology to create detailed pictures of the inside of the body. These scans are commonly used after serious injuries to help doctors assess damage to bones, blood vessels, and soft tissues. However, the long-term health risks from the radiation used in these scans—particularly the risk of developing cancer—are not well understood. In our study, we looked at long-term health data from 2662 adult trauma patients in Western Australia over a 16-year period. We found that patients who received higher doses of radiation from CT scans during their hospital stay were more likely to develop cancer and die from it later on. The most common cancer-related deaths were from lung cancer (19%), brain cancer (glioblastoma multiforme, 14%), and pancreatic cancer (14%). These findings highlight the importance of avoiding unnecessary radiation exposure, especially in younger patients who may be more vulnerable to long-term effects.

The development of trauma systems and specialized trauma services has significantly reduced trauma-related mortality and morbidity worldwide. Diagnostic imaging — particularly Computed Tomography (CT) and CT angiography — plays a critical role in the rapid assessment and management of trauma patients[1]. While nearly indispensable in modern trauma care, these imaging modalities expose patients to ionizing radiation[1–4].

The link between radiation exposure and cancer risk is widely accepted to follow a linear no-threshold (LNT) model[5], based on data from atomic bomb survivors and supported by recent epidemiological studies[6]. Large-scale research has shown that CT radiation exposure in children, adolescents, and young adults is associated with increased risks of hematological malignancies, with an excess relative risk of 1.96 per 100 mGy absorbed by

active bone marrow over a 12-year follow-up[7]. Similarly, a significant linear dose-response relationship has been observed between CT brain radiation and the risk of brain cancers[8]. However, the quantitative relationship between cumulative radiation exposure — particularly from CT scans during hospitalization for severe trauma — and subsequent cancer risk in older adults remains poorly defined[2–4]. In this study, we evaluate whether cumulative radiation exposure during the initial (index) trauma hospitalization is associated with an increased risk of developing malignant neoplasms and cancer-related mortality. Our findings demonstrate a dose-dependent association between CT-related radiation exposure and subsequent cancer incidence and mortality.

## Methods

This study is a pre-planned sub-analysis of a statewide data-linkage cohort investigating long-term health outcomes in trauma patients across Western Australia[9]. The cohort included adult patients admitted to one or more of the following trauma hospitals: Royal Perth Hospital (2004-2020), Fremantle Hospital (2011-2020), Joondalup Hospital (2011-2020), Sir Charles Gairdner Hospital (2011-2020), and Fiona Stanley Hospital (2015-2020). Royal Perth Hospital serves as Western Australia's Adult State Trauma Center and manages all adult traumatic brain and spinal injuries, as well as the majority of adult trauma cases.

Of the 29,191 trauma patients in the master cohort[9], 15,352 were admitted from 2004 onward. A subset of 2662 patients (17.3%) with available smoking data was included in the primary analysis (Fig. 1). Among these patients, 35% had traumatic brain injuries, 20% had peripheral limb injuries, 18.6% had abdominal injuries, 15.0% had thoracic injuries, and 9.2% had cervical spine injuries. Patients with a history of cancer within five years prior to or during their index trauma admissions (between January 1, 2004, and June, 1, 2020), based on public and private hospitalization records, were excluded. Radiation exposure from CT scans was quantified using data from the statewide radiology Picture Archiving and Communication System (PACS), established in 2004. Exposure was measured using dose-length product (DLP)[10]. A standard whole-body CT (WBCT) protocol may result in a DLP exceeding 3000 mGy*cm[11]. The effective dose, which better reflects the biological impact of radiation, is derived from DLP and varies by scanned body region and patient age (Table 1). Estimated effective dose data were available for 1,308 patients. However, the dataset lacked site-specific information on the body regions imaged, limiting anatomical site-specific cancer risk analysis.

Cancer diagnoses following trauma admission were identified using the International Statistical Classification of Diseases, Tenth Revision, Clinical Modification (ICD-10-CM) codes from subsequent hospitalizations. Free-text data from death certificates and cancer registries were linked to trauma records by the State Data Linkage Unit to determine causes of death. Follow-up for both cancer incidence and cancer-related mortality began on January 1, 2004, and ended on December 11, 2020, with complete outcome ascertainment for all individuals in the cohort. Results are reported in accordance with STROBE guidelines. De-identified data will be made available upon reasonable request. Ethics approval was granted by the Western Australia Department of Health. (Project Registration Number: RGS0000000404). Patient consent was waived due to observational nature of the study, the large sample size, and use of fully de-identified data.

### Statistics and Reproducibility

Data analyzed in this study — including the radiation dosage exposure, ICD-10 diagnostic codes, and Injury Severity Score (ISS) — were collected prospectively for administrative purposes. Associations between radiation dose and injury severity (measured by ISS), as well as other covariates, were assessed using Pearson's correlation coefficient ($r$). The ISS is an anatomical scoring system that quantifies injury severity by summing the squares of the highest Abbreviated Injury Scale (AIS) scores from up to three of six body regions: head, face, chest, abdomen, extremities (including pelvis), and external (e.g., burns).

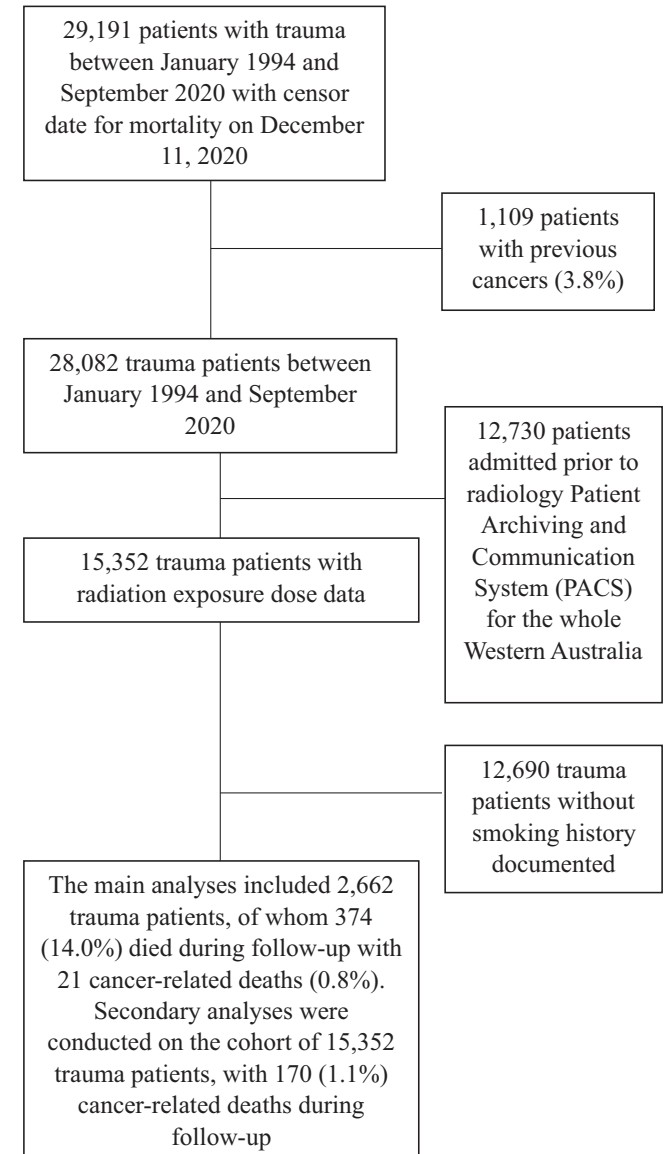

**Fig. 1 |** Flow chart showing patients included and excluded from the analyses.

Cox proportional hazards regression models were used to identify factors associated with cancer outcomes following index trauma admission. Due to the low incidence of cancer outcomes, only biologically plausible covariates — such as age, smoking status, and alcohol use (ever / never) — were included in multivariable analyses. Given the relatively low prevalence of comorbidities among younger trauma patients, parsimony models were constructed to confirm the robustness of the findings. Kaplan-Meier survival curves were used to illustrate differences in the incidence of new-onset cancers post-trauma.

Supplementary analyses included: (a) Censoring patients who died from cancer within three years of trauma; (b) Using estimated effective radiation dose instead of DLP to measure exposure; (c) Applying multiple imputation for missing smoking status; and (d) Stratifying cancer location relative to the main site of injury — presumed to be the most frequently scanned region. For each location specific analysis, patients with a trauma near the site were assigned their DLP value, while others were considered unexposed (DLP = 0). Differences in patient characteristics between smokers in the main dataset and those in the imputed datasets are described in Table S1. Notably, ISS was significantly higher among patients with recorded smoking data compared to smokers in the multiple imputation datasets. All statistical tests were 2-sided, with a significant threshold of P < .05,

**Table 1 | The estimated effective dose of radiation (in millisieverts, mSv)—which reflects the biological risk of exposure by accounting for the varying sensitivity of different organs—can be derived from the Dose-Length Product (DLP) of Computed Tomography (CT) scans for different body regions, using region-specific conversion factors. For reference, see the DLP calculator at: https://howradiologyworks.com/dlp-calculator/**

| Dose-Length-Product (mGy*cm) | Region of examination (or main region of trauma in our cohort) | Age group# | Effective Dose (mSv) |
|---|---|---|---|
| Theoretical calculations: | | | |
| 1000 | Head | Adult | 2.1 |
| 2000 | Head | Adult | 4.2 |
| *Empirical data:* | | | |
| *(N=939)* Mean 2200 (SD 1568) | Traumatic brain injury | Mean age 43 (SD 22) | *(N=525)* Mean 5.0 (SD 10.1) |
| Theoretical calculations: | | | |
| 1000 | Neck | Adult | 5.9 |
| 2000 | Neck | Adult | 11.8 |
| *Empirical data:* | | | |
| *(N=246)* Mean 1940 (SD 1338) | Cervical injury | Mean age 43 (SD 19) | *(N=108)* Mean 4.9 (SD 10.2) |
| Theoretical calculations: | | | |
| 1000 | Chest | Adult | 14.0 |
| 2000 | Chest | Adult | 28.0 |
| *Empirical data:* | | | |
| *(N=400)* Mean 1870 (SD 1342) | Thoracic injury | Mean age 48 (SD 21) | *(N=172)* Mean 6.5 (SD 13.7) |
| Theoretical calculations: | | | |
| 1000 | Abdomen and pelvis | Adult | 15.0 |
| 2000 | Abdomen and pelvis | Adult | 30.0 |
| *Empirical data:* | | | |
| *(N=495)* Mean 1990 (SD 1387) | Abdominal injury | Mean age 45 (SD 22) | *(N=206)* Mean 6.7 (SD 14.3) |

#Theoretical calculations assume effective dose increases inversely with age when patients are less than 10 years old. *SD* standard deviation.

The effective dose of radiation and DLP values for our patients, based on empirical data and stratified by their primary region of injury upon admission (assuming that region was exposed to CT radiation), are listed in the rows shaded in grey for comparison.

without adjustment for multiple comparisons. Analyses were performed using SPSS for Windows version 29.0.2.0 (IBM).

## Results

After excluding patients without smoking data (*n* = 12,690), 2662 patients (17.3%) are included in the primary analysis. The median age was 41 years (interquartile range [IQR]: 27–58), and 2016 (75.8%) were male. Alcohol use (*n* = 1325, 49.8%) and smoking (*n* = 1235, 46.4%) were common. Coexisting medical conditions were relatively uncommon: diabetes mellitus (5.3%), chronic pulmonary disease (1.8%), ischemic heart disease (1.5%), and congestive heart failure (1.5%). The median ISS was 17 (IQR: 16–22).

During a median follow-up of 5.9 years (IQR: 4.0–7.9; range: 1 day to 15.4 years), 374 patients (14.0%) died, including 21 deaths (5.6% of all deaths) attributed to malignant neoplasms. The 30-day and 1-year mortality were 2.9% and 6.4%, respectively. Patients underwent a median of 6 X-rays (IQR: 3–12) and 3 CT scans (IQR: 1–5). CT radiation exposure was below 3000 mGy*cm, the DLP of a typical whole-body scan, in 1849 patients (69.5%). The median DLP was 1941 mGy*cm (mean: 2,463; IQR: 637–3388; range: 0–26,233). The median estimated effective radiation dose was 13.0 mSv (mean: 17.0; IQR: 4.0–23.0; range: 1–237). DLP strongly correlated with the estimated effective radiation dose (*r* = 0.714, 95% confidence interval [CI]: 0.69-0.74; p = 0.001), and both were positively correlated with

ISS (Pearson *r* = 0.209 and 0.266, respectively; both p = 0.001). DLP showed weak correlations with age (*r* = −0.044; p = 0.023) and socioeconomic disadvantage index (*r* = 0.051; p = 0.012), but no significant correlation with Charlson Comorbidity Index (*r* = −0.031, p = 0.108).

Over the 16-year follow-up, 91 patients (3.4%) developed cancer, and 21 (0.8%) died from cancer-related causes. A total of 286 patients (10.8%) were exposed to DLP greater than 5000 mGy*cm — approximately one standard deviation above the mean. After censoring cancer-related deaths within three years of trauma, patients exposed to DLP > 5000 mGy·cm had significantly higher cancer-related mortality (aHR: 3.35; 95% CI: 1.20–9.38; p = 0.021) (Table 2, Fig. 2). Radiation exposure was also dose-dependently associated with increased cancer risk (aHR of 1.08 per 1000 mGy*cm increment in DLP (95%CI: 1.01–1.16; p = 0.042) (Table 3). There were no missing data for mortality outcomes or causes of death. The most common cancer-related deaths were due to lung cancers (19%), glioblastoma multiforme (14%), and pancreatic cancers (14%) (Table S2).

### Supplementary analyses

**Censoring early cancer deaths.** After excluding patients who died from cancer within three years of trauma, DLP > 5000 mGy*cm remained significantly associated with cancer-related mortality (aHR: 3.93; 95%CI: 1.20–-12.92; p = 0.024) (Table 2).

**Effective dose analysis.** The association between estimated effective radiation dose and new-onset cancer did not reach statistical significance (p = 0.081), but the trend was consistent with our primary findings (Table 3).

**Multiple imputation.** After imputing missing smoking data, the association between high radiation exposure (DLP > 5000 mGy*cm) and cancer-related mortality (aHR: 2.08; 95%CI: 1.47–2.95; p = 0.001) and cancer incidence (aHR: 1.37; 95%CI: 1.14–1.65; p = 0.001) remained robust (Tables S3-S4; Fig. S1a, b).

**Anatomical site analysis.** No significant associations were found between thoracic or abdominal cancers and corresponding injury sites. However, significant associations were observed for lymphoma/leukemia (aHR: 3.75; 95% CI: 2.01–6.99; p = 0.001) and head/neck cancers (aHR: 2.64; 95% CI: 1.14–6.11; p = 0.023) (Tables S5–S6; Figs. S2–S3).

## Discussion

Rapid assessment and treatment of trauma patients — often relatively young — frequently necessitate CT imaging, which remains indispensable in managing major trauma. To our knowledge, this is the first study to evaluate long-term cancer risks and outcomes associated with cumulative radiation exposure during initial trauma hospitalizations in older adults, where CT scans were performed for trauma-related indications rather than cancer-related symptoms.

In our cohort, short-term mortality substantially exceeded long-term cancer-related mortality. Specifically, 30-day and 1-year mortality rates were 2.9% and 6.4%, respectively, compared to a 0.8% cancer-related mortality over a 16-year follow-up. These findings are consistent with a smaller observational study[12]. However, a recent large-scale epidemiological study estimated that approximately 5% of all cancers may be attributable to diagnostic CT scans[4].

Our supplementary analyses revealed that exposure to a DLP greater than 5000 mGy*cm was significantly associated with an increased risk of hematologic malignancies, including lymphoma or leukemia (aHR 3.75; 95%CI: 2.01–6.99; p = 0.001) (Table S5), aligning with prior studies involving children, adolescents, and young adults[7]. Similarly, we observed a dose-dependent association between CT-related radiation exposure and the risk of brain and neck cancers (Tables S5 and S6)[8]. These findings support the concept of radiation-induced carcinogenesis as a stochastic process[6-8] — where the probability of cancer increases with dose, but severity does not. Consequently, minimizing unnecessary diagnostic radiation exposure remains critical, especially in younger trauma patients[13,14].

**Table 2 | Multivariable regression analysis examining the quantitative relationship between radiation exposure during the entire index trauma hospitalization and the risk of cancer-related deaths (n = 21) among 2662 trauma patients**

| Cox proportional hazards model | Hazard ratio [HR] (95% CI) | P-value |
|---|---|---|
| Age | 1.08 (1.05–1.11) per year increment | *0.001* |
| Diabetes mellitus | 1.10 (0.25–4.78) | 0.904 |
| Smoker (ever [n = 1235, 46.4% of all patients; n = 6, 28.6% of cancer–related deaths] vs never) | 1.12 (0.40–3.15) | 0.828 |
| Alcohol user (ever [n = 1325, 49.8% of all patients; n = 9, 42.9% of cancer–related deaths] vs never) | 1.12 (0.45–2.80) | 0.803 |
| CT scan radiation dose–length–product >5,000 mGy*cm during index trauma hospitalization (n = 286, 10.7% of all patients; n = 5, 23.8% of cancer–related deaths) | 3.35 (1.20–9.38) | *0.021* |
| **Cox proportional hazards model** | **Hazard ratio [HR] (95% CI)** | **P–value** |
| Age | 1.08 (1.05–1.11) per year increment | *0.001* |
| Diabetes mellitus | 1.23 (0.28–5.34) | 0.786 |
| Smoker (ever vs never) | 1.16 (0.41–3.30) | 0.776 |
| Alcohol user (ever vs never) | 1.19 (0.48–2.99) | 0.705 |
| CT scan radiation exposure during index trauma hospitalization | 1.17 (1.03–1.33) per 1000 mGy*cm increment in dose-length-product | *0.013* |
| **Cox proportional hazards model** with 7 patients who died from cancers within 3 years of trauma admission were censored | **HR (95% CI)** | **P-value** |
| Age | 1.07 (1.03–1.11) per year increment | *0.001* |
| Diabetes mellitus | 1.88 (0.40–8.71) | 0.421 |
| Smoker (ever vs never) | 0.66 (0.17–2.60) | 0.551 |
| Alcohol user (ever vs never) | 0.79 (0.25–2.47) | 0.683 |
| CT scan radiation dose-length-product >5000 mGy*cm during index trauma hospitalization | 3.93 (1.20–12.92) | *0.024* |

**Fig. 2 |** Among 2662 patients admitted for initial trauma, the incidence of cancer-related mortality was compared between those with CT scan radiation exposure (dose-length product) greater than 5000 mGy*cm and those with exposure less than 5000 mGy*cm during the entire index trauma hospitalization. Cancer-related deaths occurring within three years were censored.

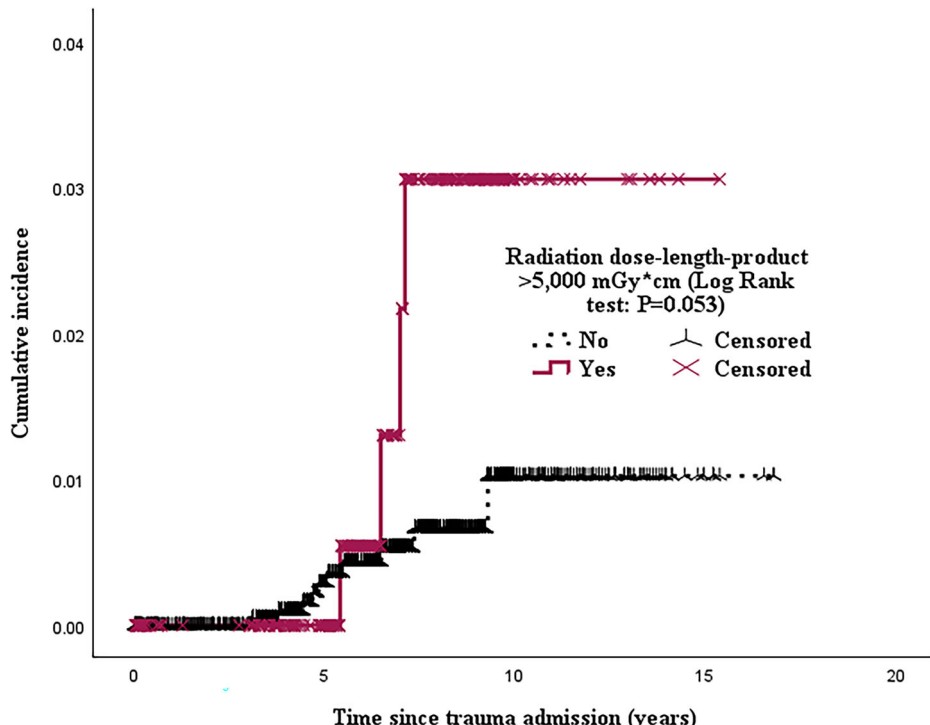

Technological advancements have demonstrated that low-dose whole-body CT (WBCT) can achieve diagnostic accuracy comparable to standard-dose WBCT while nearly halving radiation exposure[14]. This approach may reduce long-term cancer risk, although its impact on cancer outcomes in younger trauma patients remains uncertain and warrants further investigation[12].

In our primary analysis, lung cancer emerged as the most common cause of cancer-related death linked to radiation exposure, consistent with findings from a large U.S. observational study[4]. Lung cancer is also among the most frequent secondary malignancies following therapeutic radiation[15,16]. Although we could not confirm a quantitative relationship between thoracic radiation exposure and subsequent thoracic cancers in this study, clinicians should carefully weigh the immediate diagnostic benefits of CT imaging — especially when repeated — against potential long-term risks, particularly in younger patients.

**Table 3 | Multivariable regression analysis showing the quantitative relationship between radiation exposure during the entire index trauma hospitalization and the risk of new-onset cancers (n = 91) among a total of 2662 patients**

| Cox proportional hazards model | Hazard ratio (95%CI) | P-value |
|---|---|---|
| Age | 1.06 (1.05–1.07) per year increment | *0.001* |
| Ischemic heart disease | 0.68 (0.16–2.89) | 0.601 |
| Congestive heart failure | 1.00 (0.23–4.34) | 0.999 |
| Chronic pulmonary disease | 0.74 (0.18−3.05) | 0.674 |
| Diabetes mellitus | 1.14 (0.54–2.40) | 0.740 |
| Smoker (ever vs never) | 0.62 (0.36–1.06) | 0.079 |
| Alcohol user (ever vs never) | 1.67 (1.08–2.60) | *0.022* |
| CT scan radiation dose–length–product >5000 mGy*cm during index trauma hospitalization | 1.85 (1.04–3.29) | *0.038* |
| **Cox proportional hazards model** | **Hazard ratio (95%CI)** | **P-value** |
| Age | 1.06 (1.05–1.07) per year increment | *0.001* |
| Ischemic heart disease | 0.71 (0.18–3.04) | 0.648 |
| Congestive heart failure | 0.96 (0.22–4.15) | 0.952 |
| Chronic pulmonary disease | 0.74 (0.18−3.04) | 0.671 |
| Diabetes mellitus | 1.17 (0.55–2.47) | 0.683 |
| Smoker (ever vs never) | 0.62 (0.37–1.07) | 0.085 |
| Alcohol user (ever vs never) | 1.72 (1.10–2.70) | *0.017* |
| CT scan radiation dose exposure during index trauma hospitalization | 1.08 (1.01–1.16) per 1000 mGy*cm increment in dose-length-product | *0.042* |
| **Parsimony Cox proportional hazards model** # | **Hazard ratio (95%CI)** | **P-value** |
| Age | 1.06 (1.04–1.08) per year increment | *0.001* |
| Diabetes mellitus | 0.87 (0.31–2.43) | 0.795 |
| Smoker (ever vs never) | 0.60 (0.31–1.16) | 0.131 |
| Alcohol user (ever vs never) | 1.80 (1.04–3.12) | *0.036* |
| Estimated effective radiation dose (mSv) | (0.99–1.02) per mSv increment | 0.081 |

# Only 1308 patients had data on estimated effective radiation dose.

This study has several limitations. First, as an observational study, the associations may be influenced by residual confounding and do not establish causality. Second, we had substantial missing data on smoking — a known risk factor for cancer — and lacked quantitative details on this exposure. Although multiple imputation yielded consistent results, differences in injury severity between smokers in the primary dataset and imputed datasets were noted. Third, the relatively low incidence of cancer-related deaths and new-onset cancers, combined with a short median follow-up time, limited our ability to fully adjust for confounding variables. Additionally, unmeasured confounders — such as family history of cancer, occupational exposure to carcinogens, and variability in CT protocols across institutions — were not accounted for. Finally, we lacked data on the specific anatomical regions scanned, which restricted our ability to assess site-specific associations between radiation exposure and cancer risk.

## Conclusions

Radiation exposure from CT imaging during trauma hospitalization was dose-dependently associated with increased risks of new-onset cancers and cancer-related mortality. Although the immediate threat to life from trauma far outweighs the long-term cancer risk from CT exposure, our findings highlight the importance of balancing diagnostic utility with potential long-term harms. Where clinically appropriate, low-dose WBCT protocols should be considered — particularly for younger trauma patients who may be more susceptible to radiation-induced malignancies.

## Data availability

A deidentified version of the dataset can be found in Supplementary Data. Data variable definitions are given in the Supplementary Note. All additional relevant data supporting this study are available upon reasonable scientific request. Please contact the corresponding author for access.

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

## Acknowledgements
The study was funded by the Department of Health, Western Australia Merit Award for Career Development to Dr Ho (47421100).

## Author contributions
L.K.Y. and K.M.H. were responsible for ethics application, obtaining the data and drafting the manuscript. K.M.H. obtained the funding of the study and was responsible for data analysis. S.S. was responsible for linkage of radiation exposure data. All authors critically analyzed the results and agreed with the final version of the manuscript.

## Competing interests
None of the authors has any conflict of interest to declare and the decision to publish this manuscript was independent from the funder of this study.

## Additional information

[1]Department of Intensive Care, Fiona Stanley Hospital, Murdoch, WA, Australia. [2]Department of Emergency Medicine, Royal Perth Hospital, Perth, WA, Australia. [3]Department of Radiology, Royal Perth Hospital, Perth, WA, Australia. [4]Department of Anaesthesia and Intensive Care, The Chinese University of Hong Kong, Hong Kong SAR, China. [5]The Peter Hung Pain Research Institute, The Chinese University of Hong Kong, Hong Kong SAR, China. [6]The Prince of Wales Hospital, Hong Kong SAR, China. ✉e-mail: kmho@cuhk.edu.hk

