## [Transparent Peer Review file · Communications Medicine]

Dose-related association between radiation exposure from Computed Tomography (CT) scans during trauma hospitalizations and subsequent risk of developing new-onset cancers

Corresponding Author: Professor Kwok Ho

Version 0:

Reviewer comments:

Reviewer #1

(Remarks to the Author)

The study investigates the association between diagnostic CT-related radiation exposure, quantified by dose-length product (DLP), and subsequent cancer incidence and cancer-specific mortality in adult trauma patients. This topic is relevant in the context of increasing CT use in acute care, particularly among younger individuals. The focus on trauma patients, a population at high risk of repeated imaging, fills an existing gap in the literature and provides a useful complement to previous studies in pediatric and general adult cohorts.

I commend the authors for addressing a clinically important question with a carefully assembled cohort. However, while the study demonstrates methodological strengths (notably the use of PACS-derived DLP and linked cancer/mortality registries), the current version of the manuscript could benefit from several clarifications and improvements.

The Methods section lacks sufficient detail on how the final analytic sample of 2,669 patients was derived from the 9,820 patients in the master cohort. Was the exclusion of patients with cancer within 5 years the only criterion? Were there additional exclusions (missing data on smoking, alcohol, radiation dose or missing PACS data)? Please describe all inclusion and exclusion criteria explicitly, and clarify the proportion excluded due to missingness.

The process by which cancer diagnoses and causes of death were determined from "subsequent hospitalizations" and "free text" requires greater elaboration. Was the free-text search systematic? Which data sources were prioritized: hospital admissions, cancer registry, death certificates? Was outcome ascertainment complete for all individuals? Please describe how these sources were integrated and the methods used to minimize outcome misclassification.

The median follow-up time is provided, but the end date, range, maximum, and minimum follow-up durations should also be reported. This is particularly important given cancer latency. Also please clarify whether latency considerations were incorporated into the analysis (For example: exclusion of cases occurring within the first 1–2 years post-trauma). A sensitivity analysis excluding early cancer diagnoses would strengthen causal inference.

DLP is an appropriate surrogate for radiation dose, but it does not reflect organ-specific absorbed dose (can be estimated with software like NCICT), which is more biologically relevant for cancer risk. The manuscript should discuss why DLP was chosen and potential implications. Additionally, was DLP readily available for all CT scan, even or older examinations from the earlier years of the study?

The study uses data from five different hospitals over a 16-year period. CT scanner technology, protocols, and radiation outputs can vary significantly across institutions and change over time. The manuscript does not describe any methods to standardize or "harmonize" these dose measurements, which is a potential source of measurement error in the study's primary exposure variable.

Please provide the justification for selecting 5000 mGy*cm as the threshold for high radiation exposure. Was this value pre-

specified based on prior literature, or was it determined from the data distribution?

Please clarify in the Methods section how smoking and alcohol use were defined. Were these binary variables (ever/never) or were more granular data available and used?

The use of logistic regression for new-onset cancer does not account for variable follow-up durations or cancer latency, which may bias the estimated associations. Reanalysis using time-to-event methods (for example: cox regression), or sensitivity analyses excluding short follow-up cases, is recommended.

Clarification is also needed regarding covariate selection, including which variables were initially considered, how and why the included variables were chosen, and why others were excluded.

A brief explanation of what the ISS score represents would benefit readers unfamiliar with trauma metrics. Additionally, please clarify how ISS was derived, retrospectively? by trained clinicians? and whether scoring was blinded to exposure/outcome status.

Finally, the manuscript would benefit from a dedicated paragraph discussing study limitations. Suggested points include: small number of cancer-related deaths (n=21), limiting statistical power for mortality analysis, possible residual confounding (family history, occupational exposures, SES), risk of surveillance bias (more imaging leads to more incidental cancer detection), and the short median follow-up.

Reviewer #2

(Remarks to the Author)

This manuscript presents a retrospective cohort study examining the dose-response relationship between radiation exposure from CT imaging during initial trauma hospitalizations and subsequent risk of developing new-onset cancers or cancer-related mortality. Drawing from a large, statewide dataset in Western Australia spanning over 16 years, the authors analyzed 2,669 patients without prior cancers and found that higher cumulative radiation exposure (quantified by the dose-length product, DLP) was significantly associated with increased all cancer incidence and mortality. The authors interpret the association causally and conclude that there is need for careful use of CT imaging in trauma care, especially in young patients.

The manuscript represents a rather poor analysis of an otherwise possibly informative data set. There are several weaknesses in central areas of the study.

Endpoint: The endpoint is all cancer incidence and all cancer mortality. No justification is provided why such unspecific endpoints are chosen when site-specific cancer incidence is apparently available. Any causal conclusions are very difficult based on the results.

Exposure: The exposure metric is the DLP which is not commonly used in similar studies. Virtually all similar studies use absorbed organ dose. The authors should calculate the absorbed dose to selected organs and evaluate the association between absorbed dose to an organ and site-specific cancer incidence in the same organ (or statistical approaches for associating organ doses to all cancer incidence use site-specific cancer incidence as endpoint).

The flaw in the current combination of endpoint and exposure is that a subject with a CT of the abdomen and brain cancer during follow-up would contribute towards a positive association although the brain cancer cannot be caused by the CT-related radiation.

Confounding: The endpoints are known to be associated with smoking and alcohol intake. Yet, these factors are missing for a high proportion of subjects (Methods, line 76). The authors should check whether no information can perhaps be interpreted as no use. They should compare individuals with and without missing smoking and alcohol information to better understand whether missing was at random or not. They should then use appropriate methods to deal with missing data.

I missed a nuanced discussion of the study setting. There is little doubt that trauma patients benefit from imaging more than they suffer from the increased cancer risk. This should be stressed. The benefit of the particular study setting, I believe, is not to evaluate trauma patients' imaging-related cancer risk, but to select a patient group with imaging for an indication that is not associated with future cancer diagnosis. This was a limitation of the previous studies among children with CT-scans, who could have been examined for symptoms caused by an existing but undiagnosed cancer or by comorbidity of a genetic disorder which also leads to an increased cancer risk (indication bias). However, there is emerging evidence that head trauma is associated with an increased brain cancer risk. The authors therefore have to address the possibility that cancer following trauma is caused by the trauma and not by radiation exposure from imaging.

The stats section is misleading. Logistic regression is described as the method for cancer-specific survival, and Cox regression for cancer mortality. Both endpoints are survival time endpoints (not binary) and should be evaluated with Cox regression. Are the odds ratios reported actually HRs or are they just plain wrong (e.g., Table 1)?

Several factors in Table 1 have an OR (or HR?) of 0 with confidence limits from 0 to not estimable. I suspect there were no events with the respective factor present. This situation should be handled by Firth correction, which will produce meaningful estimates.

The unit of reporting the risk due to radiation exposure, per 1000 mGy*cm, does not seem to make sense. It is about one third of the dose of a typical whole body CT.

If a DLP of 3000 mGy*cm corresponds to a typical whole body CT, and the median DLP was around 2000 mGy*cm (p 6), how can the median number of CTs be 3?

There is no mention of the several large studies on the same topic. This has to be added and will show the advantage (and necessity) of using absorbed dose rather than DLP.

Figure 1: The cumulative incidence should be shown rather than the cumulative hazard, since the latter is hard to interpret. If the events of interest are cancer-related deaths, there are too many steps in the cumulative hazard curves, compared to N = 21 deaths due to cancer. Please check and correct.

Start and end of an individuals follow-up need to be specified.

I am missing a statement about the consent of the subjects to use their data.

Abstract: Mention the location of the study

Michael Hauptmann together with Luca Caramenti

Reviewer #3

(Remarks to the Author)

This succinct and well-written manuscript reports evidence that links radiation exposure from Computed Tomography to cancer incidence and mortality. After adjusting for age, several comorbidities, and other lifestyle risk factors--such as smoking and alcohol use--a statistically significant increased risk for cancer was found per 1000 mGy*cm increment of the dose-length product. The authors cautioned that the benefits of CT analyses for trauma patients should be considered within the context of the potential lifetime cancer risk for younger patients from CT exposures. Furthermore, the authors recommended additional study of low-dose whole-body CT scans for diagnostic evaluation, as there is presumably less long term risk for these exposures.

Minor comments:

Page 2: Lines 29-31: Provide a brief mention of the Australian trauma cohort

Page 3: Line 63: Cite references for linear-no-threshold model

Page 4: Use 'Cancer outcomes were assessed "with"...' to avoid redundancy

Page 5: Line 103: What is the basis for considering $P < 0.05$ as significant?

Reviewer #4

(Remarks to the Author)

I co-reviewed this manuscript with one of the reviewers who provided the listed reports. This is part of the Communications Medicine initiative to facilitate training in peer review and to provide appropriate recognition for Early Career Researchers who co-review manuscripts.

Version 1:

Reviewer comments:

Reviewer #1

(Remarks to the Author)

The authors have fully addressed my comments and incorporated the requested revisions into the manuscript. I am satisfied with the changes, and from my side no further review is required.

Reviewer #2

(Remarks to the Author)

The authors failed to sufficiently incorporate valuable comments. Several ambiguities remain in the manuscript, which need to be corrected. I am afraid that as is, the results do not add to the current evidence on the topic.

Maximum follow-up is stated as 21 years. However, the earliest start of follow-up is given as 2004 and the latest end as 2020, i.e., follow-up can maximally be 16 years.

The DICOM header of a CT scan in the PACS includes the body part. The authors would need to extract this information and use it.

The authors seem to know whether a CT was a whole body CT or not. This sheds doubt on their statement that they do not know the body part scanned.

Even if body part was not known, more could be done. E.g., the location of the injury could be used to estimate the body part scanned. Then use the data to predict the effective dose for all CTs from body part and DLP based on the 1300 CTs with DLP and effective dose.

Using multiple imputation to include 15000 subjects rather than only 2600 subjects with complete data is not a sensitivity analysis.

The results are based on 2600 of 29000 subjects, i.e., 9% of the population. Or 18% if the 15000 patients in the period 2004-2020 are used. This needs to be stated in abstract and results. The authors need to provide evidence why the results are generalizable, e.g., by showing similar distributions for variables among the 2600 included and the 15000-2600 excluded subject.

There is no binary endpoint in the study and there should therefore not be an logistic regression analysis, as 2 reviewers noted. All endpoints are time to event endpoints which should be evaluated by Cox regression. Evaluating time to event endpoints with logistic regression is inappropriate and poor statistical and epidemiological practice.

Not wanting to make causal conclusions is no excuse for presenting uninformative results. What is the point of reporting an association between radiation and cancer if one is not at least partly convinced that the association maybe causal?

The authors were recommended to use site-specific cancer as endpoints but decided to stick with all cancer. Thereby, they do not make full use of the data they have.

The linear-no-threshold model is not just based on the a-bomb data. Numerous studies since then show data which are consistent with this model.

Excluding subjects with less than 3 years follow-up introduces immortal time bias. The proper way to do this is to either censor such subjects or to start follow-up for everyone at 3 years after the trauma.

Table 1 repeats a table from an external source. The authors should rather use their own data to derive such a table and perhaps compare it with the published data.

Results from the multiple imputation analysis of 15000 subjects are not sufficiently shown. The title of sTable 3 says that 2229 new-onset cancers were included among 15447 patients. I am unsure where these data come from. The proportion of cancers per subjects is substantially higher than in the main analysis (21 in 2600).

In the description of the cohort, where the hospitals are listed, the time period of admissions should be added.

Reviewer #4

(Remarks to the Author)

I co-reviewed this manuscript with one of the reviewers who provided the listed reports. This is part of the Communications Medicine initiative to facilitate training in peer review and to provide appropriate recognition for Early Career Researchers who co-review manuscripts.

Version 2:

Reviewer comments:

Reviewer #2

(Remarks to the Author)

The authors respond to one of my comments that patients "with radiation dose documented prior to statewide implementation of the PACS radiology system" were excluded. This needs to be mentioned in the methods (with numbers).

CTs with a DLP over 3000 mGy*cm should be described as such and not as whole body CTs. It can be added that CTs with such a large DLP are likely to be whole body CTs. E.g., on page 8, "with X% of CT scans having DLP below 3000 mGy*cm, the DLP of a typical whole-body scan."

The bodypart-specific analysis using the trauma location is very interesting but needs to be described more clearly. I suggest to perform several site-specific cancer analyses (e.g.). For each analysis, use the DLP of patients with a trauma location near the site and consider all other patients as unexposed (DLP=0). If this is how the results in sTable 4 were produced, the explanation should be added to the methods. In this case, the null results for specific sites shed some doubt on an association between DLP and cancer risk. Highlighting the only site with a significant result (lymphoma and leukemia) is selective. E.g., the nonsignificant results for head/neck are not consistent with other studies. If the results were produced

differently, the table should be recalculated.

The methods section in the abstract should mention exposure metric and endpoint.

Start and end of follow-up should be described in the methods section for each endpoint (cancer incidence and cancer mortality).

The new analyses on bodypart should be mentioned in the paragraph listing additional analyses (pages 7 and 8).

What does "those who were computed to be smokers" mean?

Results for incidence and mortality should be provided for continuous (DLP in mGy*cm) and categorical exposure (>5000 mGy*cm versus <mGy*cm).

Table 1: Explain which lines are empirical data and which are theoretical calculations. Add N for empirical data.

Table 2: Add number of events for each category.

The vertical axis is labeled "Incidence" in Figure 2 and "Cumulative incidence" in Figure S1. Aren't both cumulative incidences?

Reviewer #4

(Remarks to the Author)

I co-reviewed this manuscript with one of the reviewers who provided the listed reports. This is part of the Communications Medicine initiative to facilitate training in peer review and to provide appropriate recognition for Early Career Researchers who co-review manuscripts.

Version 3:

Reviewer comments:

Reviewer #1

(Remarks to the Author)

The authors have adequately addressed the comments from Reviewer #4 and Reviewer #2. I find the revisions satisfactory and have no further major concerns.

July 15, 2025

Response to reviewers' comments: COMMSMED-25-0923

Reviewer #1 (Remarks to the Author):

The study investigates the association between diagnostic CT-related radiation exposure, quantified by dose-length product (DLP), and subsequent cancer incidence and cancer-specific mortality in adult trauma patients. This topic is relevant in the context of increasing CT use in acute care, particularly among younger individuals. The focus on trauma patients, a population at high risk of repeated imaging, fills an existing gap in the literature and provides a useful complement to previous studies in pediatric and general adult cohorts.

I commend the authors for addressing a clinically important question with a carefully assembled cohort. However, while the study demonstrates methodological strengths (notably the use of PACS-derived DLP and linked cancer/mortality registries), the current version of the manuscript could benefit from several clarifications and improvements.

The Methods section lacks sufficient detail on how the final analytic sample of 2,669 patients was derived from the 9,820 patients in the master cohort. Was the exclusion of patients with cancer within 5 years the only criterion? Were there additional exclusions (missing data on smoking, alcohol, radiation dose or missing PACS data)? Please describe all inclusion and exclusion criteria explicitly, and clarify the proportion excluded due to missingness.

We have added a new figure (Figure 1) to illustrate exclusion and inclusion of patients for the main analysis and newly added sensitivity analysis.

The process by which cancer diagnoses and causes of death were determined from “subsequent hospitalizations” and “free text” requires greater elaboration. Was the free-text search systematic? Which data sources were prioritized: hospital admissions, cancer registry, death certificates? Was outcome ascertainment complete for all individuals? Please describe how these sources were integrated and the methods used to minimize outcome misclassification.

We have expanded the section on how cancer diagnoses and causes of death were determined. The revised text is “*Cancer occurrence following trauma admission was identified using diagnostic codes from subsequent hospitalizations, based on the International Statistical Classification of Diseases, Tenth Revision, Clinical Modification (ICD-10-CM). Free-text data from death certificates and cancer registries were linked to trauma registry records by the State Data Linkage Unit to determine causes of death. The cohort was censored on December 11, 2020 with complete outcome ascertainment for all individuals.*”

The median follow-up time is provided, but the end date, range, maximum, and minimum follow-up durations should also be reported. This is particularly important given cancer latency. Also please clarify whether latency considerations were incorporated into the analysis (For example: exclusion of cases occurring within the first 1–2 years post-trauma). A sensitivity analysis excluding early cancer diagnoses would strengthen causal inference.

We have revised our text in the methods section as follows: *“Included patients had no history of cancer within five years prior to - and during - their index trauma admissions (between March 1, 2004 and June, 1, 2020), based on public and private hospitalization records. Radiation exposure from CT scans was quantified using data from the radiology Patient Archiving and Communication System (PACS), established in 2004.” “The cohort was censored on December 11, 2020 with complete outcome ascertainment for all individuals.” In the results section, “During a median follow-up of 5.9 years (IQR: 4.0-7.9; range: 1 day to 21.1 years), 374 patients (14.0%) died. Of these, 21 (5.6% of all deaths) were attributed to malignant neoplastic diseases.” In sensitivity analyses, excluding patients with < 3 years of follow-up, exposure to DLP >5,000 mGy*cm remained significantly associated with increased cancer-related mortality (HR 3.93; 95%CI: 1.20-12.92; p=0.024) (Table 2).*

DLP is an appropriate surrogate for radiation dose, but it does not reflect organ-specific absorbed dose (can be estimated with software like NCICT), which is more biologically relevant for cancer risk. The manuscript should discuss why DLP was chosen and potential implications. Additionally, was DLP readily available for all CT scan, even or older examinations from the earlier years of the study?

DLP data were available from 2004 when the Radiology Patient Archiving and Communication System (PACS) was established in 2004 in Western Australia. However, when we planned this study, we did not extract the body regions of the CT scans and hence we could not obtain the effective radiation dose (or absorbed) and also correlate site-specific cancers with the effective radiation dose. We have added a new table (Table 1) to illustrate how effective dose is related to DLP and highlighted this limitation in our discussion. *“Finally, we did not have data on the specific body regions exposed to CT scan radiation, which restricted our ability to quantify the site-specific associations between radiation exposure and cancer risk.”*

The study uses data from five different hospitals over a 16-year period. CT scanner technology, protocols, and radiation outputs can vary significantly across institutions and change over time. The manuscript does not describe any methods to standardize or "harmonize" these dose measurements, which is a potential source of measurement error in the study's primary exposure variable.

The CT protocols vary across institutions but for trauma series, the CT protocols were relatively consistent especially a majority of the patients included in this study were managed at Royal Perth Hospital. Royal Perth Hospital serves as Western Australia's Adult State Trauma Center and manages all adult traumatic brain and spinal injuries, as well as the majority of adult trauma cases. In the limitations, we revised it as *“First, as an observational study, the association between radiation exposure and cancer outcomes may be influenced by residual confounding rather than indicating a causal relationship. We had substantial missing data on smoking and alcohol use - both known risk factors for cancer - and lacked quantitative details on these exposures. Second, the low incidence of cancer-related deaths and new-onset cancers, combined with a short median follow-up time, limited our ability to fully adjust for confounding variables. In addition, unmeasured confounders - such as family history of cancers, occupational exposures to*

carcinogens, and variability in CT protocols across different institutions - were also not accounted for in our analyses.”

Please provide the justification for selecting 5,000 mGy*cm as the threshold for high radiation exposure. Was this value pre-specified based on prior literature, or was it determined from the data distribution?

*It was chosen based on the distribution of the data. We have revised our results as follows: “A total of 286 patients were exposed to DLP >5,000 mGy*cm (approximately mean + 1 standard deviation).”*

Please clarify in the Methods section how smoking and alcohol use were defined. Were these binary variables (ever/never) or were more granular data available and used?

Yes, unfortunately we only had binary nature of this information (ever vs never). This is described in the text and tables in the revised manuscript. A lack of more granular data as a limitation has been added. “We had substantial missing data on smoking and alcohol use - both known risk factors for cancer - and lacked quantitative details on these exposures.”

The use of logistic regression for new-onset cancer does not account for variable follow-up durations or cancer latency, which may bias the estimated associations. Reanalysis using time-to-event methods (for example: cox regression), or sensitivity analyses excluding short follow-up cases, is recommended.

We have presented results using both logistic regression and Cox proportional hazards models in the revised manuscript to confirm robustness of our results. In addition, a Kaplan-Meier survival analysis (with log-rank test as a revised figure – Figure 2) has been added to the revised paper. As described earlier, a sensitivity analysis excluding those with a follow-up period less than three years has been added to the revised paper.

Clarification is also needed regarding covariate selection, including which variables were initially considered, how and why the included variables were chosen, and why others were excluded.

The low incidence of cancer outcomes limited our statistical power for extensive adjustment. In the methods section, we explained this issue: “Due to the low incidence of cancer outcomes in the cohort, only biologically plausible covariates, such as age, smoking, and alcohol use (ever / never), were included in multivariable analyses. Given the relatively low prevalence of comorbidities among younger trauma patients, parsimony models were constructed to confirm the robustness of the findings.” In the discussion, we highlighted this limitation as follows: “Second, the low incidence of cancer-related deaths and new-onset cancers limited our ability to fully adjust for confounding variables and account for variability in CT protocols across different institutions.”

A brief explanation of what the ISS score represents would benefit readers unfamiliar with trauma metrics. Additionally, please clarify how ISS was derived, retrospectively? by trained clinicians? and whether scoring was blinded to exposure/outcome status.

All data used in this study were collected prospectively. We revised the methods section as follows: “Data analyzed in this study - including the radiation dosage exposure, ICD-10 diagnostic codes,

and Injury Severity Score (ISS) - were collected prospectively for administrative purposes.” “The ISS is an anatomical scoring system that quantifies injury severity by summing the squares of the highest Abbreviated Injury Scale (AIS) scores from up to three of six body regions: head, face, chest, abdomen, extremities (including pelvis), and external (e.g., burns).”

Finally, the manuscript would benefit from a dedicated paragraph discussing study limitations. Suggested points include: small number of cancer-related deaths (n=21), limiting statistical power for mortality analysis, possible residual confounding (family history, occupational exposures, SES), risk of surveillance bias (more imaging leads to more incidental cancer detection), and the short median follow-up.

The new paragraph highlighting the limitations is as follows: “This study has several limitations. First, as an observational study, the association between radiation exposure and cancer outcomes may be influenced by residual confounding rather than indicating a causal relationship. We had substantial missing data on smoking and alcohol use - both known risk factors for cancer - and lacked quantitative details on these exposures. Second, the low incidence of cancer-related deaths and new-onset cancers, combined with a short median follow-up time, limited our ability to fully adjust for confounding variables. In addition, unmeasured confounders - such as family history of cancers, occupational exposures to carcinogens, and variability in CT protocols across different institutions - were also not accounted for in our analyses. Finally, we did not have data on the specific body regions exposed to CT scan radiation, which restricted our ability to quantify the site-specific associations between radiation exposure and cancer risk.”

Reviewer #2 (Remarks to the Author):

This manuscript presents a retrospective cohort study examining the dose-response relationship between radiation exposure from CT imaging during initial trauma hospitalizations and subsequent risk of developing new-onset cancers or cancer-related mortality. Drawing from a large, statewide dataset in Western Australia spanning over 16 years, the authors analyzed 2,669 patients without prior cancers and found that higher cumulative radiation exposure (quantified by the dose-length product, DLP) was significantly associated with increased all cancer incidence and mortality. The authors interpret the association causally and conclude that there is need for careful use of CT imaging in trauma care, especially in young patients.

The manuscript represents a rather poor analysis of an otherwise possibly informative data set. There are several weaknesses in central areas of the study.

Endpoint: The endpoint is all cancer incidence and all cancer mortality. No justification is provided why such unspecific endpoints are chosen when site-specific cancer incidence is apparently available. Any causal conclusions are very difficult based on the results. Exposure: The exposure metric is the DLP which is not commonly used in similar studies. Virtually all similar studies use absorbed organ dose. The authors should calculate the absorbed dose to selected organs and evaluate the association between absorbed dose to an organ and site-specific cancer incidence in

the same organ (or statistical approaches for associating organ doses to all cancer incidence use site-specific cancer incidence as endpoint).

We agreed with our reviewer that we cannot make causal conclusions based on our results and have highlighted this in our discussion of the limitations of the study. *“First, as an observational study, the association between radiation exposure and cancer outcomes may be influenced by residual confounding rather than reflecting a causal relationship.”* We were also limited by a lack of site-specific data on the body regions of the CT imaging in our dataset and hence could not correlate site-specific cancers and radiation effective dose at the corresponding regions. *“Finally, we did not have data on the specific body regions exposed to CT scan radiation, which restricted our ability to assess absorbed radiation doses and quantify site-specific cancer risks.”* In the methods section, we have added the following statements: *“The effective dose, which better reflects the biological impact of radiation, is derived from the DLP and varies depending on the scanned body region and patient age (Table 1). In this study, the estimated effective dose was available only in 1,311 patients. In addition, our dataset did not include site-specific information on the body regions imaged by CT, so we were not able to assess the site-specific association between radiation exposure and cancer outcomes.”*

The flaw in the current combination of endpoint and exposure is that a subject with a CT of the abdomen and brain cancer during follow-up would contribute towards a positive association although the brain cancer cannot be caused by the CT-related radiation.

We agreed with our reviewer that without granular level of data, a definitive causative conclusion cannot be made. We have added this limitation to our discussion. *“First, as an observational study, the association between radiation exposure and cancer outcomes may be influenced by residual confounding rather than indicating a causal relationship. We had substantial missing data on smoking and alcohol use - both known risk factors for cancer - and lacked quantitative details on these exposures.”* Nonetheless, the totality of evidence from other studies in children and young adults including those published by our reviewer would support the hypothesis that diagnostic radiation especially at high doses may not completely safe. These important references have been added and highlighted in the introduction of the revised manuscript.

Confounding: The endpoints are known to be associated with smoking and alcohol intake. Yet, these factors are missing for a high proportion of subjects (Methods, line 76). The authors should check whether no information can perhaps be interpreted as no use. They should compare individuals with and without missing smoking and alcohol information to better understand whether missing was at random or not. They should then use appropriate methods to deal with missing data.

We agreed that missing data was an important limitation. We have added a sensitivity analysis using multiple imputation and the results (reported in the Supplementary materials) were consistent.

I missed a nuanced discussion of the study setting. There is little doubt that trauma patients benefit from imaging more than they suffer from the increased cancer risk. This should be stressed. The benefit of the particular study setting, I believe, is not to evaluate trauma patients' imaging-related

cancer risk, but to select a patient group with imaging for an indication that is not associated with future cancer diagnosis. This was a limitation of the previous studies among children with CT-scans, who could have been examined for symptoms caused by an existing but undiagnosed cancer or by comorbidity of a genetic disorder which also leads to an increased cancer risk (indication bias). However, there is emerging evidence that head trauma is associated with an increased brain cancer risk. The authors therefore have to address the possibility that cancer following trauma is caused by the trauma and not by radiation exposure from imaging.

We have highlighted that all CT scans were done for trauma management reasons for the patients and not for cancer related symptoms. We agreed with our reviewer that trauma could lead to an increased risk of developing cancers, but such association has less biological plausibility based on existing literature than radiation leading to an increased risk of cancer outcomes.

The stats section is misleading. Logistic regression is described as the method for cancer-specific survival, and Cox regression for cancer mortality. Both endpoints are survival time endpoints (not binary) and should be evaluated with Cox regression. Are the odds ratios reported actually HRs or are they just plain wrong (e.g., Table 1)?

We have presented both logistic and Cox regression analyses in our revised manuscript to confirm robustness of our results.

Several factors in Table 1 have an OR (or HR?) of 0 with confidence limits from 0 to not estimable. I suspect there were no events with the respective factor present. This situation should be handled by Firth correction, which will produce meaningful estimates.

Thank you for this excellent suggestion. We have used Firth correction in our logistic regression in the revised manuscript.

The unit of reporting the risk due to radiation exposure, per 1000 mGy*cm, does not seem to make sense. It is about one third of the dose of a typical whole body CT. If a DLP of 3000 mGy*cm corresponds to a typical whole body CT, and the median DLP was around 2000 mGy*cm (p 6), how can the median number of CTs be 3?

We have clarified this because most patients did not have a whole body CT scan. "Patients underwent a median of 6 X-rays (IQR: 3-12) and 3 CT scans (IQR: 1-5), with most CT scans not being whole-body scans."

There is no mention of the several large studies on the same topic. This has to be added and will show the advantage (and necessity) of using absorbed dose rather than DLP.

We have added two important large-scale epidemiological studies on risks of diagnostic radiation in children and young adults. Bosch de Basea, M., et al. Risk of hematological malignancies from CT radiation exposure in children, adolescents and young adults. *Nat Med.* 29, 3111-3119 (2023). Hauptmann, M., et al. Brain cancer after radiation exposure from CT examinations of children and young adults: results from the EPI-CT cohort study. *Lancet Oncol.* 24, 45-53 (2023). In discussing the limitations, this specific limitation is highlighted in the revised manuscript. "Finally, we did

not have data on the specific body regions exposed to CT scan radiation, which restricted our ability to assess absorbed radiation doses and quantify site-specific cancer risks.”

Figure 1: The cumulative incidence should be shown rather than the cumulative hazard, since the latter is hard to interpret. If the events of interest are cancer-related deaths, there are too many steps in the cumulative hazard curves, compared to N = 21 deaths due to cancer. Please check and correct.

We have revised this figure to be Figure 2 reporting incidence (unadjusted) of new-onset cancers by Kaplan-Meier curves.

Start and end of an individuals follow-up need to be specified.

The start and end date of admission and censored date have been added.

I am missing a statement about the consent of the subjects to use their data.

A new statement has been added about patient consent. *“Patient consent was waived due to observational nature of the study, the large sample size, and the use of fully deidentified data.”*

Abstract: Mention the location of the study

The methods section of the abstract has been revised as *“This statewide cohort study utilized data from all public and private hospitalizations, death records, and cancer registries in Western Australia”*

Reviewer #3 (Remarks to the Author):

This succinct and well-written manuscript reports evidence that links radiation exposure from Computed Tomography to cancer incidence and mortality. After adjusting for age, several comorbidities, and other lifestyle risk factors--such as smoking and alcohol use--a statistically significant increased risk for cancer was found per 1000 mGy*cm increment of the dose-length product. The authors cautioned that the benefits of CT analyses for trauma patients should be considered within the context of the potential lifetime cancer risk for younger patients from CT exposures. Furthermore, the authors recommended additional study of low-dose whole-body CT scans for diagnostic evaluation, as there is presumably less long term risk for these exposures.

Minor comments:

Page 2: Lines 29-31: Provide a brief mention of the Australian trauma cohort

We have added the details of our cohort in the methods section as follows: *“This study is a pre-planned sub-analysis of a statewide data-linkage cohort investigating long-term health outcomes in trauma patients across Western Australia.⁸ The cohort included patients aged 17 years and*

older who were admitted to one or more of the following trauma hospitals: Royal Perth Hospital, Fremantle Hospital, Joondalup Hospital, Sir Charles Gairdner Hospital, and Fiona Stanley Hospital. Royal Perth Hospital serves as Western Australia's Adult State Trauma Center and manages all adult traumatic brain and spinal injuries, as well as the majority of adult trauma cases.

Of the 29,191 trauma patients in the master cohort, a subset of 2,669 patients with complete covariate data was analyzed initially (Figure 1). Among these, 35% had traumatic brain injuries, 20% had peripheral limb injuries, 18.6% had abdominal injuries, 15.0% had thoracic injuries, and 9.2% had cervical spine injuries.”

Page 3: Line 63: Cite references for linear-no-threshold model

The following reference has been added: “Shah, D.J., Sachs, R.K., Wilson, D.J. Radiation-induced cancer: a modern view. *Br J Radiol.* 85, e1166-73 (2012).”

Page 4: Use ‘Cancer outcomes were assessed “with”...’ to avoid redundancy

Where appropriate, we have use cancer outcomes as a generic term to reduce redundancy.

Page 5: Line 103: What is the basis for considering $P < 0.05$ as significant?

With the low incidence of cancer outcomes and limited sample size and median follow-up period, we did not have the statistical power to allow us in using a more stringent p-value, such as < 0.01 , in this study.

Reviewer #4 (Remarks to the Author):

I co-reviewed this manuscript with one of the reviewers who provided the listed reports. This is part of the Communications Medicine initiative to facilitate training in peer review and to provide appropriate recognition for Early Career Researchers who co-review manuscripts.

Thank you so much for your comments and time to review our work.

Cover letter describing responses to reviewers on submission COMMSMED-25-0923A

Reviewers' comments:

Reviewer #1 (Remarks to the Author):

The authors have fully addressed my comments and incorporated the requested revisions into the manuscript. I am satisfied with the changes, and from my side no further review is required.

Thank you for confirming that we had responded adequately to your comments earlier.

Reviewer #2 (Remarks to the Author):

The authors failed to sufficiently incorporate valuable comments. Several ambiguities remain in the manuscript, which need to be corrected. I am afraid that as is, the results do not add to the current evidence on the topic.

We have further revised our manuscript and I hope our study findings may strengthen the idea that radiation exposure from CT imaging is not completely risk free in relationship to development of cancers.

Maximum follow-up is stated as 21 years. However, the earliest start of follow-up is given as 2004 and the latest end as 2020, i.e., follow-up can maximally be 16 years.

I apologized for this oversight. Some patients did have radiation dose exposure documented prior the statewide implementation of the PACS radiology system in 2004. For consistent purposes, these patients had been removed from our analyses. The maximum follow-up period was 17 years (from Jan 1, 2004 to December 2020).

The DICOM header of a CT scan in the PACS includes the body part. The authors would need to extract this information and use it.

We understand that this information was available on our PACS but unfortunately, when the data was planned and data extraction was conducted, that information was not requested and hence our datasets do not have data on the anatomical site of the CT imaging.

The authors seem to know whether a CT was a whole body CT or not. This sheds doubt on their statement that they do not know the body part scanned.

That statement was a reflection of the fact that many patients did not have a radiation exposure dose that is compatible with a whole-body CT (~3000 mGy*cm).

Even if body part was not known, more could be done. E.g., the location of the injury could be used to estimate the body part scanned. Then use the data to predict the effective dose for all CTs from body part and DLP based on the 1300 CTs with DLP and effective dose.

Thank you for this advice and I have added a new sTable 4, illustrating the relationship between anatomical site of injury and cancers. With the exception of lymphoma or leukemia (aHR 3.75; 95%CI: 2.01-6.99; p=0.001), we did not find a consistent association between cancer types and the anatomical site of injury – presumed to be the region most frequently scanned (sTable 4).

This could be due to either inadequate statistical power in the stratified analyses or the imprecise nature of the assumption that the anatomical site of injury was the same site of CT imaging.

Using multiple imputation to include 15000 subjects rather than only 2600 subjects with complete data is not a sensitivity analysis.

We have amended this term as one of the supplementary analyses.

The results are based on 2600 of 29000 subjects, i.e., 9% of the population. Or 18% if the 15000 patients in the period 2004-2020 are used. This needs to be stated in abstract and results. The authors need to provide evidence why the results are generalizable, e.g., by showing similar distributions for variables among the 2600 included and the 15000-2600 excluded subject.

We have added this information to the abstract and results, and a new sTable 1 has been added. ISS was indeed higher among smokers in the primary dataset compared to imputed smokers. We have added this limitation to our discussion.

There is no binary endpoint in the study and there should therefore not be an logistic regression analysis, as 2 reviewers noted. All endpoints are time to event endpoints which should be evaluated by Cox regression. Evaluating time to event endpoints with logistic regression is inappropriate and poor statistical and epidemiological practice.

We have removed all logistic regression analyses in the latest version of the manuscript.

Not wanting to make causal conclusions is no excuse for presenting uninformative results. What is the point of reporting an association between radiation and cancer if one is not at least partly convinced that the association maybe causal?

Association in a dose-related fashion strengthens its causal relationship. We believe our findings are consistent with earlier studies, including the effects of radiation exposure from CT imaging on risk of leukemia or lymphoma.

The authors were recommended to use site-specific cancer as endpoints but decided to stick with all cancer. Thereby, they do not make full use of the data they have.

We have added sTable 4 to describe the site-specific cancer risk in relation to site of injury, with a precautionary note that there was an assumption that site of injury was the same site of CT imaging.

The linear-no-threshold model is not just based on the a-bomb data. Numerous studies since then show data which are consistent with this model.

A new reference has been added to broaden the support for the LNT model.

Excluding subjects with less than 3 years follow-up introduces immortal time bias. The proper way to do this is to either censor such subjects or to start follow-up for everyone at 3 years after the trauma.

We have revised it as censoring those who died from cancers within three years of trauma and results remained unchanged.

Table 1 repeats a table from an external source. The authors should rather use their own data to derive such a table and perhaps compare it with the published data.

We have added our local data to Table 1, also with a precautionary note for the assumption that site of injury was the site of CT imaging and hence radiation exposure.

Results from the multiple imputation analysis of 15000 subjects are not sufficiently shown. The title of sTable 3 says that 2229 new-onset cancers were included among 15447 patients. I am unsure where these data come from. The proportion of cancers per subjects is substantially higher than in the main analysis (21 in 2600).

This error (due to duplicated count from the imputed datasets) has been corrected. The analysis included a total of 591 new-onset cancer cases among 15,352 patients using the imputed datasets.

In the description of the cohort, where the hospitals are listed, the time period of admissions should be added.

This information has been added.

Reviewer #4 (Remarks to the Author):

I co-reviewed this manuscript with one of the reviewers who provided the listed reports. This is part of the Communications Medicine initiative to facilitate training in peer review and to provide appropriate recognition for Early Career Researchers who co-review manuscripts.

Thank you for your time in reviewing our work.

Cover letter describing responses to reviewers on submission COMMSMED-25-0923B

Reviewers' comments:

Reviewer #2 (Remarks to the Author):

The authors respond to one of my comments that patients "with radiation dose documented prior to statewide implementation of the PACS radiology system" were excluded. This needs to be mentioned in the methods (with numbers).

Seven patients in the core dataset who had radiation exposure data documented were admitted prior to January 1, 2004. These patients were removed in our repeated analyses. The slight reduction in patient number has not made significant impact on our results, including the total number of cancer deaths (n=21) and incidence (n=91). Changes are highlighted in yellow in text and also Figure 1 in the revised manuscript.

CTs with a DLP over 3000 mGy*cm should be described as such and not as whole body CTs. It can be added that CTs with such a large DLP are likely to be whole body CTs. E.g., on page 8, "with X% of CT scans having DLP below 3000 mGy*cm, the DLP of a typical whole-body scan."

The revised text reads as "*CT radiation exposure was below 3,000 mGy*cm, the DLP of a typical whole-body scan, in 1,849 patients (69.5%).*"

The bodypart-specific analysis using the trauma location is very interesting but needs to be described more clearly. I suggest to perform several site-specific cancer analyses (e.g.). For each analysis, use the DLP of patients with a trauma location near the site and consider all other patients as unexposed (DLP=0). If this is how the results in sTable 4 were produced, the explanation should be added to the methods. In this case, the null results for specific sites shed some doubt on an association between DLP and cancer risk. Highlighting the only site with a significant result (lymphoma and leukemia) is selective. E.g., the nonsignificant results for head/neck are not consistent with other studies. If the results were produced differently, the table should be recalculated.

Thank you for pointing this important point out. We did a restricted analysis previously which reduced the statistical power of the analysis. On the advice of our reviewer, we have reanalyzed the data and found that both blood cancers and head/neck cancers were associated with the radiation dose exposure. These results are consistent with two recent publications cited as reference [7] and [8] in the revised manuscript: Bosch de Basea, M., et al. Risk of hematological malignancies from CT radiation exposure in children, adolescents and young adults. *Nat Med.* 29, 3111-3119 (2023). Hauptmann, M., et al. Brain cancer after radiation exposure from CT examinations of children and young adults: results from the EPI-CT cohort study. *Lancet Oncol.* 24, 45-53 (2023).

The methods section in the abstract should mention exposure metric and endpoint.

The revised text now reads as "*We conducted a statewide cohort study to examine the relationship between CT-related radiation exposure —measured by dose-length-product (DLP)*

—and cancer outcomes among adult trauma patients in Western Australia from 2004 to 2020. Patients with a documented cancer diagnosis within five years prior to trauma were excluded.”

Start and end of follow-up should be described in the methods section for each endpoint (cancer incidence and cancer mortality).

The revised manuscript now reads as “*Follow-up for both cancer incidence and cancer-related mortality began on January 1, 2004, and ended on December 11, 2020, with complete outcome ascertainment for all individuals in the cohort.*”

The new analyses on bodypart should be mentioned in the paragraph listing additional analyses (pages 7 and 8).

The revised manuscript now reads as “*and (d) Stratifying cancer location relative to the main site of injury — presumed to be the most frequently scanned region. For each location specific analysis, patients with a trauma near the site were assigned their DLP value, while others were considered unexposed (DLP=0).*”

What does “those who were computed to be smokers” mean?

We have revised this statement to improve clarity. The revised text now reads as “*Differences in patient characteristics between smokers in the main dataset and those in the imputed datasets are described in Table S1.*”

Results for incidence and mortality should be provided for continuous (DLP in mGy*cm) and categorical exposure (>5000 mGy*cm versus <mGy*cm).

We have added these results to the existing Tables and added a new Table S6 for the site-specific relationship to radiation exposure as a continuous predictor.

Table 1: Explain which lines are empirical data and which are theoretical calculations. Add N for empirical data.

We have added the N for empirical data related to DLP and effective radiation dose (mSv) and reformatted the Table to improve clarity.

Table 2: Add number of events for each category.

Table 2 reports the results of the multivariable analyses for predictors of cancer-related mortality. We assume our reviewer is interested to know the ‘number of events’ in relation to the risk factors. The left column of the Table has been expanded to report these numbers as follows:

Smoker (ever [n=1235, 46.4% of all patients; n=6, 28.6% of cancer-related deaths] vs never)

Alcohol user (ever [n=1325, 49.8% of all patients; n=9, 42.9% of cancer-related deaths] vs never)

CT scan radiation dose-length-product >5,000 mGy*cm during index trauma hospitalization (n=286, 10.7% of all patients; n=5, 23.8% of cancer-related deaths)

The vertical axis is labeled "Incidence" in Figure 2 and "Cumulative incidence" in Figure S1. Aren't both cumulative incidences?

Yes, our reviewer was correct. It was cumulative incidence. We also identified that the legend was incorrect and it should be cancer-related mortality as the outcome of interest. I have redrawn the graph after censoring the patients who died within three years after trauma.

Reviewer #4 (Remarks to the Author):

I co-reviewed this manuscript with one of the reviewers who provided the listed reports. This is part of the Communications Medicine initiative to facilitate training in peer review and to provide appropriate recognition for Early Career Researchers who co-review manuscripts.

Thank you for your time and effort in improving our work, resulting in an improved manuscript with additional details.